# How frequent is routine use of probiotics in UK neonatal units?

Neaha Patel ,[1] NeoTRIPS Collaborative Group, Katie Evans ,[2] Janet Berrington,[3,4] Lisa Szatkowski ,[5] Kate Costeloe,[1,6] Shalini Ojha ,[5,7] Paul Fleming ,[1,6] Cheryl Battersby [2]

PF and CB contributed equally.

## ABSTRACT

**Objective** There is a lack of UK guidance regarding routine use of probiotics in preterm infants to prevent necrotising enterocolitis, late-onset sepsis and death. As practices can vary, we aimed to determine the current usage of probiotics within neonatal units in the UK.

**Design and setting** Using NeoTRIPS, a trainee-led neonatal research network, an online survey was disseminated to neonatal units of all service levels within England, Scotland, Northern Ireland and Wales in 2022. Trainees were requested to complete one survey per unit regarding routine probiotic administration.

**Results** 161 of 188 (86%) neonatal units responded to the survey. 70 of 161 (44%) respondents routinely give probiotics to preterm infants. 45 of 70 (64%) use the probiotic product *Lactobacillus acidophilus* NCFM/*Bifidobacterium bifidum* Bb-06/*B. infantis* Bi-26 (Labinic™). 57 of 70 (81%) start probiotics in infants ≤32 weeks' gestation. 33 of 70 (47%) had microbiology departments that were aware of the use of probiotics and 64 of 70 (91%) had a guideline available. Commencing enteral feeds was a prerequisite to starting probiotics in 62 of 70 (89%) units. The majority would stop probiotics if enteral feeds were withheld (59 of 70; 84%) or if the infant was being treated for necrotising enterocolitis (69 of 70; 99%). 24 of 91 (26%) units that did not use probiotics at the time of the survey were planning to introduce them within the next 12 months.

**Conclusions** More than 40% of all UK neonatal units that responded are now routinely administering probiotics, with variability in the product used. With increased probiotic usage in recent years, there is a need to establish whether this translates to improved clinical outcomes.

## INTRODUCTION

Routine administration of probiotics to prevent necrotising enterocolitis (NEC), late-onset sepsis (LOS) and death in preterm infants remains contentious. Evidence of benefit from large randomised controlled trials (RCTs), together with data from pre-implementation and post-implementation studies, is inconsistent.[1–4] The most recent Cochrane meta-analysis stated that probiotics *may* reduce the risk of NEC (relative risk [RR] 0.54, 95% CI 0.45 to 0.65 (54 trials, 10 604 infants; I²=17%)) and *probably* reduce mortality (RR 0.76, 95% CI 0.65 to 0.89; (51

**Correspondence to**
Dr Cheryl Battersby; c.battersby@imperial.ac.uk

### WHAT IS ALREADY KNOWN ON THIS TOPIC

⇒ A 2018 survey of probiotic use in the UK reported that 17% of tertiary-level neonatal intensive care units were routinely using probiotics.

⇒ There are inconsistencies in the body of literature as to whether probiotics can prevent necrotising enterocolitis, late-onset sepsis and death.

⇒ Many trials evaluating probiotics in preterm infants have used different strains, contributing to a lack of consensus on which product to use.

### WHAT THIS STUDY ADDS

⇒ Over 40% of responding UK neonatal units and 62% of responding tertiary-level neonatal intensive care units currently use probiotics.

⇒ There is heterogeneity in the probiotic product used.

⇒ The increase in use offers the opportunity for further national research reviewing the impact of probiotics use on clinical outcomes.

### HOW THIS STUDY MIGHT AFFECT RESEARCH, PRACTICE OR POLICY

⇒ This survey may inform future studies that evaluate efficacy of routine probiotic administration.

trials, 10 170 infants; I²=0%)) and late-onset invasive infection (RR 0.89, 95% CI 0.82 to 0.97; (47 trials, 9762 infants; I²=19%)). This review showed limited evidence of benefit for infants <1000g and recommended further assessment of probiotics in RCTs so long as families and caregivers supported such a study.[5]

Historical uncertainties around optimal probiotic strains for use in preterm infants and of probiotics safety have likely contributed to a lower uptake of their use.[6] A large network meta-analysis comprehensively evaluated efficacy of different probiotic strains and found that some may have more evidence of benefit than others.[7] The same review cautions that without clear evidence of efficacy for some probiotics, 'clinicians may be left using inadequately tested, potentially unsafe and possibly ineffective treatments'. A 2018 telephone survey (with a 100% response

rate) of 58 neonatal intensive care units (NICUs) providing tertiary-level care in the UK revealed that 17% of centres routinely administered probiotics.[8] In 2020, the European Society for Paediatric Gastroenterology Hepatology and Nutrition (ESPGHAN) made conditional recommendations for certain probiotic strains in preterm infants but cautioned that these were made with low certainty of benefit.[9] Parent-advocacy groups support probiotic use.[10]

In light of the most recent Cochrane review and ESPGHAN recommendations, it is likely that more UK neonatal units are moving to routine probiotic administration despite uncertainties around evidence of benefit. In this study, we aimed to identify current practice on probiotic use in UK neonatal units using the Neonatal Trainee Led Research and Improvement Projects (NeoTRIPS) network, a UK national neonatal trainee-led research initiative.

## METHODS

A 21-item exploratory survey instrument (online supplemental material) was developed by the research team in May 2022 to determine the current use of probiotics in UK neonatal units. The structure was designed in 'Google Forms',[11] in which respondents were asked three questions to acquire basic neonatal unit information, including the neonatal Operational Delivery Network (ODN) location and unit care level (tick-box responses) and the name of the neonatal unit (free-text response). After this, based on a *yes/no* answer to the following screening question 'does your neonatal unit currently use probiotics?', participants were directed to one of two different sets of questions.

Participants responding 'no' to the initial screening question were directed to two further multiple-choice questions asking if they intended to implement routine probiotic use within the next 12 months and whether parents had made enquiries about probiotics. 'Yes' respondents were steered to 15 additional questions to ascertain unit probiotic administration practices. These consisted of 12 multiple-choice style questions, with 4 containing an optional free-text response and 3 open-ended questions. These questions captured the following information: which probiotic product was being used; probiotic initiation criteria (based on gestational age, birth weight and enteral feeds); post-menstrual gestation at which probiotics are discontinued; contraindications to probiotic use; presence of a unit-specific probiotics guideline and parent information leaflets; microbiology department awareness of neonatal probiotic use; and probiotic administration during scenarios when an infant had sepsis, NEC and/or was made nil by mouth. The survey was trialled by the research team which comprised of a medical statistician, five neonatal consultants and two senior neonatal trainees and underwent five iterations before the content was finalised. The research team aimed for a minimum 70% response rate to the survey based on similar targets set for other UK neonatal clinical practice surveys.[12]

The project was introduced at the 2022 Summer meeting of the NeoTRIPS collaborative group. NeoTRIPS is a national, trainee-led, neonatal research network affiliated with the British Association of Perinatal Medicine.[13] Regional NeoTRIPS leads for each neonatal ODN were provided with the survey link, which was then disseminated to local NeoTRIPS leads for each participating neonatal unit. The survey invitation contained instructions for one survey to be completed per unit by a trainee. While trainee seniority was not captured in the survey, participants were encouraged to discuss responses they were uncertain of (such as whether microbiology were aware) with senior unit clinicians. Survey reminders were sent to the regional NeoTRIPS leads fortnightly during the survey period.

There are 15 neonatal ODNs including 188 units in the UK. Centres caring for newborn infants in the UK are stratified into three levels: level 1 which refers to special care units (SCUs); level 2 or local neonatal units (LNUs); and highly specialised level 3 NICUs.[14]

The survey results were collated in a Microsoft Excel file and data were summarised using descriptive statistics for quantitative responses.

### Patient and public involvement

In August 2022, focus group sessions with parents, former NICU patients and advocacy groups recruited through the neoWONDER collaborative[15] were held to establish perspectives on the use of routinely collected neonatal data to monitor probiotic efficacy during routine administration. It was explained to participants that the survey report could be instrumental in designing any future studies and the members of this group felt this was important. The focus group was not involved in the design of this survey of clinicians' clinical practice.

## RESULTS
### Survey response rate and probiotic use

Between 13 August 2022 and 31 October 2022, 161 of 188 (86%) neonatal units completed the survey. There was a NICU response rate of 50 of 57 (88%), LNU response rate of 72 of 83 (87%) and SCU response rate of 39 of 48 (81%). Seventy of 161 (44%) responding units currently use probiotics, constituting 31 of 50 (62%), 28 of 72 (39%) and 11 of 39 (28%) of responding NICUs, LNUs and SCUs, respectively. Figure 1 outlines the response rate and probiotic use by unit designation.

### Survey responses for units routinely using probiotics

Table 1 outlines survey responses from the 70 units that confirmed routine probiotic use.

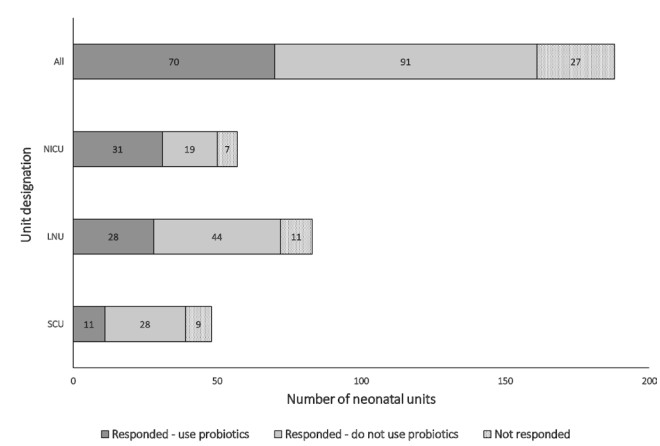

**Figure 1** : **Histogram of response rates based on neonatal unit level.** LNU, local neonatal unit; NICU, neonatal intensive care unit; SCU, special care unit.

### Additional responses from units using probiotics
#### Birth weight
Fifty-nine of 70 (84%) units using probiotics additionally had birth weight as part of their probiotic administration criteria, with 56 of 59 (95%) starting probiotics in infants ≤1500 g; 1 of 59 (2%) in infants ≤1600 g; 1 of 59 (2%) in infants ≤1800 g; and 1 of 59 (2%) not specifying a birth weight criterion.

#### When to stop probiotics
The majority of units (55 of 70 (79%)) would discontinue probiotics at 34 weeks' postmenstrual age, with 1 of 70 (1%) discontinuing at 32 weeks; 2 of 70 (3%) at 35 weeks; 2 of 70 (3%) at 36 weeks; and 2 of 70 (3%) at term. Three of 70 (4%) units would stop probiotics at discharge regardless of postmenstrual age. Five of 70 (7%) units would discontinue probiotics at either 34 weeks' postmenstrual age or discharge, depending on whether they were commenced due to gestational age or birth weight, respectively.

#### Contraindications to using probiotics
Thirty-one of 70 (44%) units listed contraindications to starting probiotics, and where free-text comments were provided, these included: intestinal atresia, NEC, sepsis and/or suspected immunocompromise.

#### Probiotic use during abdominal concerns including NEC
If enteral feeds are stopped, 59 of 70 (84%) units would also stop probiotics; 9 of 70 (13%) would continue, and 2 of 70 (3%) would reduce or modify the dose.

Sixty-nine of 70 (99%) units would stop probiotics completely for infants being treated for NEC, and just one unit stated that they would continue. Probiotics would be restarted after an episode of NEC once enteral feeds are reintroduced and/or a set volume of feed is tolerated in 53 of 69 (77%) units. In 11 of 69 (16%) units, they would only be restarted based on a consultant and/or surgical team decision. One unit stated they would not restart probiotics after NEC, one would only restart

**Table 1** Survey results for units using probiotics (n=70)

| Survey question | n (%) |
|---|---|
| **Probiotic choice** | |
| *Lactobacillus acidophilus* NCFM/*Bifidobacterium bifidum* Bb-06/*B. infantis* Bi-26 (Labinic™) | 45 (64) |
| *B. infantis* Bb-02/*B. lactis* Bb-12/*Streptococcus thermophilus* (ProPrems®) | 15 (21) |
| *B. bifidum/L. acidophilus* (Infloran®) | 10 (14) |
| **Gestation cut-off for probiotic administration** | |
| ≤34 weeks | 10 (14) |
| ≤33 weeks | 2 (3) |
| ≤32 weeks | 57 (81) |
| ≤28 weeks | 1 (1) |
| **When probiotics are started** | |
| After enteral feeds are commenced | 62 (89) |
| Soon after birth | 8 (11) |
| **Guideline for probiotic administration** | |
| Yes | 64 (91) |
| No | 4 (6) |
| Unknown | 2 (3) |
| **Microbiology team awareness of neonatal probiotic use** | |
| Yes | 33 (47) |
| No | 2 (3) |
| Unknown | 35 (50) |
| **Probiotic use during antibiotic administration** | |
| Continue | 55 (79) |
| Stop | 15 (21) |
| Reduce or modify | 0 (0) |
| **Parent information leaflets** | |
| Yes | 32 (46) |
| No | 30 (43) |
| Unknown | 8 (11) |
| **Parental refusal of probiotics** | |
| Yes | 5 (7) |
| No | 65 (93) |

once antibiotics (for NEC) are stopped and one would only restart once the baby was repatriated to their local hospital from the surgical centre. Two of 69 (3%) units were unsure regarding their policy of restarting probiotics after an episode of NEC.

### Survey responses for units not using probiotics
Table 2 outlines survey responses from the 91 units that do not routinely use probiotics.

### DISCUSSION
This up-to-date survey with a high response rate represents national practice relating to probiotic use in UK

**Table 2** Survey results for units not using probiotics (n=91)

| Survey question | n (%) |
|---|---|
| Intention to introduce routine probiotics within the next 12 months | |
| Yes | 24 (26) |
| No | 67 (74) |
| Parents enquiring regarding probiotic use | |
| Yes | 18 (20) |
| No | 73 (80) |

neonatal units. The survey confirms a notable increase in routine probiotic administration, especially in NICUs (now 62%), compared with the last survey in 2018 when this figure was 17%.[8] This may be due to the recent position paper from ESPGHAN recommending specific probiotic strains to reduce rates of NEC (despite low certainty of benefit),[9] which may in turn have increased clinician confidence in using probiotic interventions.

The most frequently used probiotic product in UK neonatal units at the time the survey was conducted was a combination of *Lactobacillus acidophilus, Bifidobacterium bifidum* Bb-06 and *B. infantis* Bi-26 (Labinic™). While this probiotic has not been extensively evaluated in large RCTs,[16] it has been reported to show efficacy against NEC, LOS and unadjusted mortality in some studies.[3] Furthermore, it does not meet current ESPGHAN guidelines. The guidelines caution against using some strains of *L. acidophilus* due to insufficient safety data relating to the effects of D-lactate production by certain strains of this bacteria.[9]

The second most commonly used combination is *B. infantis, B. lactis* Bb-12 and *Streptococcus thermophiles* (ProPrems®). This combination of probiotic bacteria has been evaluated in two RCTs, but neither was powered for NEC as a primary outcome.[1 17] The combination is however endorsed in the ESPGHAN position paper and further RCTs of its efficacy against NEC and mortality in extremely preterm babies are planned (ClinicalTrials.gov: NCT05604846).

Choosing which probiotic to use on a neonatal unit can sometimes be dependent on availability rather than using products which have robust evidence of benefit. Van den Akker *et al* cautioned against this, stating that 'clinicians may be left using inadequately tested, potentially unsafe and possibly ineffective treatments'.[7] Furthermore, there is inconsistency in the evidence of benefit in pre-implementation and post-implementation studies using Labinic™ and the combination of *B. bifidum* and *L. acidophilus* (Infloran®) in UK neonatal units. While one study reported a significant NEC reduction during epochs after these probiotics were routinely implemented,[3] another more recent study did not show evidence of benefit.[4] Why these differences exist in different geographical locations but within the same country is unclear, but may reflect differences in infants, unit practices or disease categorisation.

Inconsistencies in probiotic efficacy are also recognised across large RCTs. Among the two largest, the ProPrems trial reported a significant reduction in NEC among infants randomised to a probiotic combination containing *B. infantis, S. thermophilus* and *B. lactis*[1]; however, for participants in the PiPS trial, there was no evidence of NEC reduction among infants randomised to *B. breve* BBG-001.[2] Neither trial reported significant reductions in LOS or mortality. Strain selection and probiotic combinations are therefore important and may act differently in different populations. Not all are effective, and none has been shown to definitively reduce all of NEC, LOS and death.

Probiotic safety is frequently highlighted when considering routine probiotic supplementation in preterm infants. Adverse clinical events have mostly taken the form of isolated case reports of bacteraemias and given the large numbers of preterm babies exposed to probiotics in clinical trials, these reports appear infrequently.[18] Probiotic contamination has also been reported and associated with death from fungal contaminants.[19] Using probiotic products that are manufactured to the highest pharmaceutical standard is thus imperative. In this current survey, more than 50% of respondents were unsure if their local microbiology department was aware that probiotics were being used on the neonatal unit. This may represent the responses of more junior members of the team or a failure to educate clinicians about the importance of probiotic implementation surveillance. It also potentially highlights the lack of appreciation/awareness of the importance of involving local laboratory services to ensure there is a system in place to detect adverse events associated with probiotic use. When using live interventions, it is imperative that unit guidelines are designed with input from local microbiology teams so that patient and product safety can be monitored.[20] If they are not, adverse events associated with probiotic use could be under-reported.

Over half of responding units stated they do not currently use probiotics, but we did not specifically ask why this was. Lack of availability of products with clear evidence of efficacy, conflicting reports from large RCTs, cost, a lack of national consensus guidelines and safety concerns are recognised to contribute to clinician reluctance to implement routine probiotic use. Among units not currently using probiotics, 26% (24 units) indicated they intend to start routine probiotic administration within the next year. With the increasing number of units moving towards routine administration, it is important that probiotic efficacy is monitored systematically across the UK's neonatal population. One way this could be achieved is to monitor efficacy using large population databases of routinely collected electronic patient record data, such as the National Neonatal Research Database. To do this, these databases would need to record probiotic use in a manner that allows easy extraction of these data and reliable comparisons of clinical endpoints in babies exposed to probiotics versus those who have not

been exposed. During patient and parent focus groups associated with designing this project, participants were unanimously supportive towards maximising the utility of routinely collected data to address some of the uncertainties around probiotic efficacy. Monitoring safety during routine implementation is equally important and using systems such as the British Paediatric Surveillance Unit[21] could be one way to capture and report any adverse events associated with probiotic use.

One of the greatest strengths of this study was the high response rate of 86%. This exceeded the target of 70% and demonstrates the willingness of trainees to undertake research. Furthermore, the survey was completed within 12 weeks making results available in a timely and efficient manner. The main limitation of the study was that parts of the survey that allowed free-text entries were not always completed (eg, contraindications to using probiotics).

## CONCLUSIONS

This survey confirms that an increasing number of UK neonatal units are administering probiotics to preterm babies. As units move towards routine probiotic use, systems to monitor efficacy are needed as some of the probiotics being used have limited evidence of benefit. Evaluation of efficacy could be achieved by interrogating routinely collected national datasets. Robust reporting of any adverse outcomes associated with probiotic use is also needed so that adequate safety monitoring occurs in parallel with routine implementation.

**Author affiliations**
[1]Department of Neonatology, Homerton Healthcare NHS Foundation Trust, London, UK
[2]Neonatal Medicine, School of Public Health Faculty of Medicine, Imperial College London, London, UK
[3]Department of Neonatology, Newcastle upon Tyne Hospitals NHS Foundation Trust, Newcastle upon Tyne, UK
[4]Translational and Clinical Research Institute, Newcastle University, Newcastle upon Tyne, UK
[5]Centre for Perinatal Research, Lifespan and Population Health, School of Medicine, University of Nottingham Faculty of Medicine and Health Sciences, Nottingham, UK
[6]Genomics and Child Health, Queen Mary University of London Barts and The London School of Medicine and Dentistry, London, UK
[7]Neonatal Unit, University Hospitals of Derby and Burton NHS Foundation Trust, Derby, UK

**Acknowledgements** We would like to thank all the neonatal units that contributed to this survey.

**Collaborators** NeoTRIPS Collaborative Group: Steve Abbey, Bushra Abdul-Malik, Abdulhakim Abdurrazaq, Arameh Aghababaie, Shreya Agrawal, Saud Ahmed, Faith Akano, Muhammad Rehan Akhtar, Oghenetekevwe Patrick Akpofure, Myriam Segovia Almiron, Namita Anand, Jessica Archibald, Harriet Aughey, Lynnlette Aung, Thandi Aung, Moataz Badawy, Pramila Bade, Mary-Rose Ballard, Naomi Bell, Shreesh Bhat, Andrada Maria Bianu, Catherine Black, Gennie Booth, Karla Buerano, Nuala Calder, Claire Caldwell, Chinnu Chandran, Shavin Chellen, Nathan Collicott, Lizaveta Collins, Ruth Cousins, Brandy Cox, Deborah Davidson, Leanne Dearman, Rajkumar Dhandayuthapani, Shweta Dixit, Kouros Driscoll, Alshaimaa Eldeeb, Teim Eyo, Jessica Farnan, Yasin Fatine, Michelle Fernandes, Lauren Ferretti, Poppy Flanagan, Eileen Foster, Caroline Fraser, Christopher Freeman, Natasha G,

Neelakshi Ghosh, Abhrajit Giri, Sion Glaze, Amy Grant, Kirti Gupta, Fergus Harnden, Saqib Hasan, Craig Haverstock, Jayne Hillier, Benjamin Holter, Alison Hopper, Kate Hooper, Zoe Howard, Rachel Hutchinson, Shana Irvine, Mais Ismail, Matilda Iverson, Camilla James, Sam Jay, Katie Jenkins, Allan Jenkinson, Swati Jha, Manohar Joishy, Rhiannon Jones, Mia Kahvo, Carla Kantyka, Caroline Kargbo, Almutassem Kazkaz, Shelley Knights, Nikoletta Kottarakou, Carianne Lewis, Christine Lim, Naomi Lin, Helen Lloyd, Catherine Longley, Spandana Rupa Madabhushi, Carys Mangan, Diane McCarter, Joe McConville, Aodhan McGillian, Tasneem Modan, Ahmed Mohamed, Evangelia Myttaraki, Rhianna Netherton, Maria Orford, Tal Oryan, Joanna O'Sullivan, Niha Peshimam, Jennifer Peterson, Salil Pradhan, Patrycja Prusak, Ayesha Rahim, Daniel Ratnaraj, Rosie Roots, Afza Sadiq, Emilie Seager, Naveed Shahzad, Adwa Shalabi, Naseem Sharif, Rebecca Smith, James Stevens, Claire Strauss, Jane Sundarsingh, Jonathan Talbot, Jasmine Taylor, Katie Taylor, Justinas Teiserskas, Sumit Thankur, Toby Thenat, Chibuko Ukeje, Alice Unsworth, Carl Van Heyningen, Elena Raluka Vlad, Anna Waghorn, Emma Williams, Lucinda Winckworth, Aneurin Young.

**Contributors** NP, KE, JB, LS, KC, SO, PF and CB were involved in the conception, methodology, writing, reviewing and editing of the manuscript. NP, KE and the NeoTRIPS Collaborative Group were involved in data acquisition. CB is acting as guarantor.

**Funding** NeoTRIPS Collaborative Group supported by CW+ Charity and the Chelsea and Westminster Hospital NHS Foundation Trust, London, UK.

**Competing interests** None declared.

**Patient and public involvement** Patients and/or the public were involved in the design, or conduct, or reporting, or dissemination plans of this research. Refer to the Methods section for further details.

**Patient consent for publication** Not required.

**Ethics approval** Formal ethical approval was not required for this evaluation of service provision. No patient-identifiable information was sought or gathered in this survey.

**Provenance and peer review** Not commissioned; externally peer reviewed.

**Data availability statement** All data relevant to the study are included in the article or uploaded as supplemental information.

**ORCID iDs**
Neaha Patel http://orcid.org/0000-0002-9577-8223
Katie Evans http://orcid.org/0000-0002-0503-9807
Lisa Szatkowski http://orcid.org/0000-0003-3295-5891
Shalini Ojha http://orcid.org/0000-0001-5668-4227
Paul Fleming http://orcid.org/0000-0001-6027-4212
Cheryl Battersby http://orcid.org/0000-0002-2898-553X

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
