## [Reviewer comments · BMJ Paediatrics Open]

ARTICLE DETAILS

TITLE (PROVISIONAL)	How Frequent is Routine Use of Probiotics in UK Neonatal Units?
AUTHORS	Patel, Neaha Collaborative Group, NeoTRIPS Evans, Katie Berrington, Janet Szatkowski, Lisa Costeloe, Kate Ojha, Shalini Fleming, Paul Battersby, Cheryl

VERSION 1 - REVIEW

REVIEWER	Ayoub Mitha
REVIEW RETURNED	30-Apr-2023

GENERAL COMMENTS	Thank you for the opportunity to review your manuscript. In this paper, the authors sought to determine the current usage of probiotics within neonatal units in the United Kingdom in 2022. The context of the study is well described, regarding the previous survey in UK with 17% of NICUs routinely administered probiotics in 2018. Following the ESPGHAN position paper in 2020, probiotic use is currently a hot topic in neonatology. Study design and methodology The survey design is well explained and detailed, with an aim of minimum 70% response rate based upon similar targets set for other UK neonatal clinical practice surveys. I appreciate the paragraph regarding the neoWONDER collaborative. I would have suggested a question regarding the reason when neonatal unit is not using probiotics. Interpretation Well written, and discussed. The most frequently used probiotic in UK neonatal units: combination of Lactobacillus acidophilus, Bifidobacterium bifidum Bb-06 and Bifidobacterium infantis Bi-26 does not meet current ESPGHAN guidelines as it is well written in the discussion. Authors may explain further ESPGHAN position such as "L. acidophilus is a partially D-lactate producing strain for which there is insufficient safety data available in preterm infants." Furthermore, authors may detail that the second most commonly used combination B. infantis, B. lactis Bb-12 and Streptococcus thermophiles is one of the strains recommended by ESPGHAN guidelines. Recent pre-post implementation study using this
--

	combination in preterm infants 28-31 weeks (Nutrients 2022, 14(17), 3646; https://doi.org/10.3390/nu14173646) supports data that probiotics seemed associated with less morbidities, but also improved feeding tolerance and growth, less x-rays, and shorter time on antibiotics. To complete the knowledge gap for probiotics supplementation among extremely preterm infants, a multicenter RCT is currently on going: Probiotic Supplementation (B. infantis, B. lactis Bb-12 and Streptococcus thermophiles) in Extremely Preterm Infants in Scandinavia (PEPS) (ClinicalTrials.gov Identifier: NCT05604846) Authors could discuss that regarding the increasing number of UK neonatal units using routinely probiotics supplementation, national guidelines might be the next step in UK, to help clinicians to harmonize practices such as Canadian (Canadian Paediatric Society. Position statement: using probiotics in the paediatric population. Updated Dec 9, 2022. Accessed April 22, 2023) and Swedish guidelines (Abrahamsson, T.; Ahlsson, F.; Diderholm, B.; Elfvin, A.; Pupp, H.; Wackernagel, D.; Domellöf, M. Probiotika under Neonatalperioden. 2020. Available online: https://neo.barnlakarforeningen.se/wp-content/uploads/sites/14/2020/02/Probiotika-neo-va%CC%8Ardprogram-20020205.pdf (accessed on 21 April 2023). I thank Sara Bjurman for her help in completing this review.
--	--

REVIEWER	Dr. Elaine M. Boyle Univ Leicester
REVIEW RETURNED	04-May-2023

GENERAL COMMENTS	Thank you for the opportunity to review your work. It is great to see papers resulting from the work of trainee networks. The use of probiotics in preterm infants is controversial, with limited and conflicting evidence from robust randomised controlled trials, Evidence of benefit has been more convincing in numerous studies of term born infants. Nevertheless, potential gains for preterm infants in terms of NEC and sepsis prevention are important and this has led to probiotics being introduced in many neonatal units, in spite of theoretical (and some reported cases) risks of sepsis from the organisms administered. To my knowledge, there has not been a recent survey of practice, so this is useful information to gauge the degree of enthusiasm for probiotics in the United Kingdom, and current practice. The response rate is higher than many surveys and this likely reflects the wish for trainees to support their colleagues in this project. I have a few minor queries and comments: The questionnaires were disseminated by the NeoTRIPS Group via NeoTRIPS leads for each ODN and unit. It is not clear to me how, at that point, the participants were selected. Were they in general, the NeoTRIPS leads, or other trainees. If only one response is required from each neonatal unit, were any checks made to ensure that the relevant information was available to the person completing the
---

	questionnaire? It is likely that understanding and knowledge of practice may vary between trainees with different levels of neonatal experience. Did you collect any information on experience/seniority of the participants? I would like to see this clarified. How might this have affected your results? In your abstract you state that 70/161 (44%) units were using probiotics. I could not see this in the main body of the paper in the results section. This should be added for context. I think it would also be helpful to see, within the text, a brief breakdown of the responses by type of unit - this would support the chart you show in Figure 1. I am interested to see that you chose to include "parent refusal of probiotics". Did you explore whether the use of probiotics is routinely discussed with parents, or whether some form of consent is obtained? In some of the questions, there are a number of "unknown" responses. This is particularly the case regarding "microbiology team awareness". You suggest that this may be due to the fact that this may represent responses from junior members of the team. If you have information regarding seniority of respondents, then perhaps you could check to see if this is the case. If not, then I think it should be explicitly discussed as a limitation. This could also be contributing to the absence of free text responses when requested. In your conclusion, you state that your survey confirms that an increasing number of units are using probiotics. However, it also shows that more than 50% of responding units have not adopted this into their practice, as well as a lack of consistency in the way in which they are being used. It would be nice to see some reflection on this finding and speculation as to why this might be. Does it, for example, simply reflect the current limitations of the current evidence base and clinicians' reluctance to introduce something that is non-evidence based, or other reasons (cost implications etc). General comments: There are some minor typographical errors that require correction.
--	---

REVIEWER	Dr. Ann Hickey Children's Health Ireland at Temple Street, Neonatology
REVIEW RETURNED	05-May-2023

GENERAL COMMENTS	This manuscript aimed to establish levels of routine probiotic usage in preterm infants on neonatal units across the UK. The authors successfully achieved their aim with 161/188 neonatal units responding to their trainee led survey. The impressive response rate suggests that trainee led research networks like NEOTRIPs may be an effective way of reaching multiple sites and encouraging participation in studies. The paper is well written and easy to understand with clear explanations of the methodology and results. The results are interesting and confirm increasing routine use of probiotics as well as highlighting the ongoing problem of inconsistent practice where evidence remains uncertain. It would be useful to understand if unit guidelines were devised independently in smaller (and bigger) units or whether there were
---

	regional variations ie was network practice a factor in guiding local unit practice? If so it might suggest greater possibilities of unifying practice and enabling further studies to try to establish clearer evidence of efficacy. Highlighting of the role of microbiology when using probiotics is important and possibly not widely appreciated. It is noteworthy that 50% of the trainee responders answered 'don't know'. Discussion and education around probiotic use in neonatal training is common but may not always include an emphasis on microbiology and safety aspects. It would be interesting to understand more about the units who are not planning to introduce probiotics and why? There acknowledgment that parent advocacy groups support probiotic use may make this a difficult position going forward. Although the potential for adverse incidents with probiotic use is mentioned – the survey did not ask that question. Do the authors think that would be useful or that such information may be over or underreported? The last line of page 15 stating that 'these reports appear less frequently' could perhaps be clarified. Overall this is a useful and engaging paper which adds to the existing literature and reinforces the need for ongoing research in this already widely studied topic to identify clearer evidence to guide practice for clinicians and infants.
--	--

VERSION 1 – AUTHOR RESPONSE

Reviewer: 1

1. Thank you for the opportunity to review your manuscript. In this paper, the authors sought to determine the current usage of probiotics within neonatal units in the United Kingdom in 2022. The context of the study is well described, regarding the previous survey in UK with 17% of NICUs routinely administered probiotics in 2018. Following the ESPGHAN position paper in 2020, probiotic use is currently a hot topic in neonatology.

Study design and methodology

The survey design is well explained and detailed, with an aim of minimum 70% response rate based upon similar targets set for other UK neonatal clinical practice surveys.

I appreciate the paragraph regarding the neoWONDER collaborative.

Response P1: Thank you for this feedback.

2. I would have suggested a question regarding the reason when neonatal unit is not using probiotics.

Response P2: We are very grateful for this suggestion and should we undertake future probiotic surveys, we will definitely consider including questions around why units may not routinely administer them.

3. The most frequently used probiotic in UK neonatal units: combination of Lactobacillus acidophilus, Bifidobacterium bifidum Bb-06 and Bifidobacterium infantis Bi-26 does not meet current ESPGHAN guidelines as it is well written in the discussion. Authors may explain further ESPGHAN position such as "L. acidophilus is a partially D-lactate producing strain for which there is insufficient safety data available in preterm infants."

4. Furthermore, authors may detail that the second most commonly used combination B. infantis, B. lactis Bb-12 and Streptococcus thermophiles is one of the strains recommended by ESPGHAN guidelines. Recent pre-post implementation study using this combination in preterm infants 28-31 weeks (Nutrients 2022, 14(17), 3646;

<https://gbr01.safelinks.protection.outlook.com/?url=https%3A%2F%2Fdoi.org%2F10.3390%2Fnu14173646&data=05%7C01%7Cneaha.patel%40nhs.net%7C29f308954f4146cb120608db4d9bf81f%7C37c354b285b047f5b22207b48d774ee3%7C0%7C1%7C638189104501346258%7CUnknown%7CTWFpbGZsb3d8eyJWIjoiMC4wLjAwMDAiLCJQIjoiV2luMzliLCJBTiI6Ik1haWwiLCJXVCi6Mn0%3D%7C3000%7C%7C%7C&sdata=thKrg2NPd5RkWeFX9kq9nkFCvwCHFsjNuv0vqOMcHcg%3D&reserved=0>) supports data that probiotics seemed associated with less morbidities, but also improved feeding tolerance and growth, less x-rays, and shorter time on antibiotics.

5. To complete the knowledge gap for probiotics supplementation among extremely preterm infants, a multicenter RCT is currently on going:

Probiotic Supplementation (B. infantis, B. lactis Bb-12 and Streptococcus thermophiles) in Extremely Preterm Infants in Scandinavia (PEPS) (ClinicalTrials.gov Identifier: NCT05604846)

Response P3, P4 & P5: Thank you for these really helpful suggestions. We have modified two paragraphs in the discussion to encompass these points but have not included the paper by Mitha et al as we have not reported selective secondary outcomes from other trials. The paragraphs now read as follows:

'The most frequently used probiotic product in UK neonatal units at the time the survey was conducted was a combination of Lactobacillus acidophilus, Bifidobacterium bifidum Bb-06 and Bifidobacterium infantis Bi-26 (Labinic™). Whilst this probiotic has not been extensively evaluated in large RCTs [16], it has been reported to show efficacy against NEC, LOS and unadjusted mortality in some studies [3]. Furthermore, it does not meet current ESPGHAN guidelines. The guidelines caution against using some strains of Lactobacillus acidophilus due to insufficient safety data relating to the effects of D-lactate production by certain strains of this bacteria [9].

The second most commonly used combination is Bifidobacterium infantis, Bifidobacterium lactis Bb-12 and Streptococcus thermophiles (ProPrems®). This combination of probiotic bacteria has been evaluated in two RCTs, but neither was powered for NEC as a primary outcome [1, 17]. The combination is however endorsed in the ESPGHAN position paper and further RCTs of its efficacy against NEC and mortality in extremely preterm babies are planned (ClinicalTrials.gov: NCT05604846)'.

6. Authors could discuss that regarding the increasing number of UK neonatal units using routinely probiotics supplementation, national guidelines might be the next step in UK, to help clinicians to harmonize practices such as Canadian (Canadian Paediatric Society. Position statement: using probiotics in the paediatric population. Updated Dec 9, 2022. Accessed April 22, 2023) and Swedish guidelines (Abrahamsson, T.; Ahlsson, F.; Diderholm, B.; Elfvin, A.; Pupp, H.; Wackernagel, D.; Domellöf, M. Probiotika under Neonatalperioden. 2020. Available online: <https://gbr01.safelinks.protection.outlook.com/?url=https%3A%2F%2Fneo.barnlakarforeningen.se%2Fwp-content%2Fuploads%2Fsites%2F14%2F2020%2F02%2FProbiotika-neo-va%25CC%258Ardprogram-20020205.pdf&data=05%7C01%7Cneaha.patel%40nhs.net%7C29f308954f4146cb120608db4d9bf81f%7C37c354b285b047f5b22207b48d774ee3%7C0%7C1%7C638189104501346258%7CUnknown%7CTWFpbGZsb3d8eyJWIjoiMC4wLjAwMDAiLCJQIjoiV2luMzliLCJBTiI6Ik1haWwiLCJXVCi6Mn0%3D%7C3000%7C%7C%7C&sdata=Xe%2B4tRFXTRYHbXCmruoTLCCJ8bw%2BHjfhqP7ntqwNY%3D&reserved=0> (accessed on 21 April 2023).

Response P6: All three reviewers asked for more reflection on the reasons behind why, more than half of units do not currently use probiotics. To address this, we have included additional reflection on why this may be the case. It is important to note however, that when it comes to national guidelines, given the inconsistencies in evidence of benefit, it is not the unanimous opinion of the authors that regional or national guidelines would be beneficial based on the level of evidence available at this time. The additional reflections are included in the following paragraph:

'Over half of responding units stated they do not currently use probiotics but we did not specifically ask why this was. Lack of availability of products with clear evidence of efficacy, conflicting reports from large RCTs, cost, a lack of national consensus guidelines and safety concerns are recognised to contribute to clinician reluctance to implement routine probiotic use. Amongst units not currently using probiotics, 26% (24 units) indicated they intend to start routine probiotic administration within the next year. With the increasing number of units moving towards routine administration, it is important that probiotic efficacy is monitored systematically across the UK's neonatal population. One way this could be achieved is to monitor efficacy using large population databases of routinely-collected electronic patient record data, such as the National Neonatal Research Database (NNRD).'

Reviewer: 2

7. Thank you for the opportunity to review your work. It is great to see papers resulting from the work of trainee networks. The use of probiotics in preterm infants is controversial, with limited and conflicting evidence from robust randomised controlled trials, Evidence of benefit has been more convincing in numerous studies of term born infants. Nevertheless, potential gains for preterm infants in terms of NEC and sepsis prevention are important and this has led to probiotics being introduced in many neonatal units, in spite of theoretical (and some reported cases) risks of sepsis from the organisms administered. To my knowledge, there has not been a recent survey of practice, so this is useful information to gauge the degree of enthusiasm for probiotics in the United Kingdom, and current practice. The response rate is higher than many surveys and this likely reflects the wish for trainees to support their colleagues in this project.

Response P7: Thank you for this feedback.

8. The questionnaires were disseminated by the NeoTRIPS Group via NeoTRIPS leads for each ODN and unit. It is not clear to me how, at that point, the participants were selected. Were they in general, the NeoTRIPS leads, or other trainees. If only one response is required from each neonatal unit, were any checks made to ensure that the relevant information was available to the person completing the questionnaire? It is likely that understanding and knowledge of practice may vary between trainees with different levels of neonatal experience. Did you collect any information on experience/seniority of the participants? I would like to see this clarified. How might this have affected your results?

Response P8: We have modified the paragraph in the methods to now include this additional information. The paragraph reads as follows:

'The project was introduced at the 2022 Summer meeting of the of the NeoTRIPS collaborative group. NeoTRIPS is a national, trainee-led, neonatal research network affiliated with the British Association of Perinatal Medicine (BAPM) [13]. Regional NeoTRIPS leads for each neonatal ODN were provided with the survey link which was then disseminated to local NeoTRIPS leads for each participating neonatal unit. The survey invitation contained instructions for one survey to be completed per unit by a trainee. Whilst trainee seniority was not captured in the survey, participants were encouraged to discuss responses they were uncertain of (such as whether microbiology were aware) with senior unit clinicians. Survey reminders were sent to the regional NeoTRIPS leads fortnightly during the survey period'.

9. In your abstract you state that 70/161 (44%) units were using probiotics. I could not see this in the main body of the paper in the results section. This should be added for context. I think it would also be helpful to see, within the text, a brief breakdown of the responses by type of unit - this would support the chart you show in Figure 1.

Response P9: The opening paragraph of the results has been modified to include these data and now reads as follows:

'Between 13th August 2022 and 31st October 2022, 161/188 (86%) neonatal units completed the survey. There was a NICU response rate of 50/57 (88%), LNU response rate of 72/83 (87%) and SCU response rate of 39/48 (81%). 70/161 (44%) responding units currently use probiotics, constituting 31/50 (62%), 28/72 (39%) and 11/39 (28%) of responding NICUs, LNUs and SCUs, respectively. Figure 1 outlines the response rate and probiotic use by unit designation'.

10. I am interested to see that you chose to include "parent refusal of probiotics". Did you explore whether the use of probiotics is routinely discussed with parents, or whether some form of consent is obtained?

Response P10: Unfortunately, we did not capture any additional information on this question other than the binary answer which was reported. We will definitely consider including more in-depth questions around parental opinions on probiotics in any future surveys including any consent which an individual unit might require.

11. In some of the questions, there are a number of "unknown" responses. This is particularly the case regarding "microbiology team awareness". You suggest that this may be due to the fact that this may represent responses from junior members of the team. If you have information regarding seniority of respondents, then perhaps you could check to see if this is the case. If not, then I think it should be explicitly discussed as a limitation. This could also be contributing to the absence of free text responses when requested.

Response P11: As in response P8, trainees completing the survey were encouraged to discuss it with senior unit clinicians, but we did not implicitly ask whether or not they did. The comment was speculative and reviewer three (Dr Hickey) has commented that it may also reflect that adequate training and education about the importance of surveillance may not have filtered through. We hope that the additional clarity in the methods section addressed in P8 will partly address this comment but have also modified the paragraph in the discussion to now read as follows:

'In this current survey, more than 50% of respondents were unsure if their local microbiology department was aware that probiotics were being used on the neonatal unit. This may represent the responses of more junior members of the team or indeed a failure to properly educate clinicians about the importance of probiotic implementation surveillance. It also potentially highlights the lack of appreciation/awareness of the importance of involving local laboratory services to ensure there is a system in place to detect adverse events associated with probiotic use'.

12. In your conclusion, you state that your survey confirms that an increasing number of units are using probiotics. However, it also shows that more than 50% of responding units have not adopted this into their practice, as well as a lack of consistency in the way in which they are being used. It would be nice to see some reflection on this finding and speculation as to why this might be. Does it, for example, simply reflect the current limitations of the current evidence base and clinicians' reluctance to introduce something that is non-evidence based, or other reasons (cost implications etc).

Response P12: We have modified the following paragraph in the discussion to reflect these important points. Similar points were raised by the other reviewers (see reviewer 1 P6).

'Over half of responding units stated they do not currently use probiotics but we did not specifically ask why this was. Lack of availability of products with clear evidence of efficacy, conflicting reports from large RCTs, cost, a lack of national consensus guidelines and safety concerns are recognised to contribute to clinician reluctance to implement routine probiotic use. Amongst units not currently using probiotics, 26% (24 units) indicated they intend to start routine probiotic administration within the next year. With the increasing number of units moving towards routine administration, it is important that probiotic efficacy is monitored systematically across the UK's neonatal population. One way this could be achieved is to monitor efficacy using large population databases of routinely-collected electronic patient record data, such as the National Neonatal Research Database (NNRD).'

Reviewer: 3

13. Dr. Ann Hickey, Children's Health Ireland at Temple Street, Children's Health Ireland at Crumlin

This manuscript aimed to establish levels of routine probiotic usage in preterm infants on neonatal units across the UK. The authors successfully achieved their aim with 161/188 neonatal units responding to their trainee led survey. The impressive response rate suggests that trainee led research networks like NEOTRIPs may be an effective way of reaching multiple sites and encouraging participation in studies. The paper is well written and easy to understand with clear explanations of the methodology and results. The results are interesting and confirm increasing routine use of probiotics as well as highlighting the ongoing problem of inconsistent practice where evidence remains uncertain.

Response P13: Thank you for this feedback.

14. It would be useful to understand if unit guidelines were devised independently in smaller (and bigger) units or whether there were regional variations ie was network practice a factor in guiding local unit practice? If so it might suggest greater possibilities of unifying practice and enabling further studies to try to establish clearer evidence of efficacy.

15. It would be interesting to understand more about the units who are not planning to introduce probiotics and why? There acknowledgment that parent advocacy groups support probiotic use may make this a difficult position going forward.

Response P14 & 15: Unfortunately, we did not capture this level of granularity but we fully agree these are important points to consider and should be incorporated into any future surveys. All three reviewers asked for more reflection on the reasons behind why more than half of units do not currently use probiotics and to address this, we have included additional reflection on why this may be the case. It is important to note however, that when it comes to guidelines, given the inconsistencies in evidence of benefit, it is not the unanimous opinion of the authors that regional or national guidelines would be beneficial based on the level of evidence available at this time. The additional reflections are included in the following paragraph:

'Over half of responding units stated they do not currently use probiotics but we did not specifically ask why this was. Lack of availability of products with clear evidence of efficacy, conflicting reports from large RCTs, cost, a lack of national consensus guidelines and safety concerns are recognised to contribute to clinician reluctance to implement routine probiotic use. Amongst units not currently using probiotics, 26% (24 units) indicated they intend to start routine probiotic administration within the next year. With the increasing number of units moving towards routine administration, it is important that probiotic efficacy is monitored systematically across the UK's neonatal population. One way this could be achieved is to monitor efficacy using large population databases of routinely-collected electronic patient record data, such as the National Neonatal Research Database (NNRD).'

16. Highlighting of the role of microbiology when using probiotics is important and possibly not widely appreciated. It is noteworthy that 50% of the trainee responders answered 'don't know'. Discussion and education around probiotic use in neonatal training is common but may not always include an emphasis on microbiology and safety aspects.

17. Although the potential for adverse incidents with probiotic use is mentioned – the survey did not ask that question. Do the authors think that would be useful or that such information may be over or underreported? The last line of page 15 stating that 'these reports appear less frequently' could perhaps be clarified.

Response P16 & 17: We are really grateful for this observation and highlighting that we may have simply not stressed the importance of surveillance during training is a really valid point. We have tried to address both points by correcting a paragraph in the discussion which now reads as follows:

'Probiotic safety is frequently highlighted when considering routine probiotic supplementation in preterm infants. Adverse clinical events have mostly taken the form of isolated case reports of bacteraemias and given the large numbers of preterm babies exposed to probiotics in clinical trials, these reports appear infrequently [18]. Probiotic contamination has also been reported and associated with death from fungal contaminants [19]. Using probiotic products that are manufactured to the highest pharmaceutical standard is thus imperative. In this current survey, more than 50% of respondents were unsure if their local microbiology department was aware that probiotics were being used on the neonatal unit. This may represent the responses of more junior members of the team or a failure to educate clinicians about the importance of probiotic implementation surveillance. It also potentially highlights the lack of appreciation/awareness of the importance of involving local laboratory services to ensure there is a system in place to detect adverse events associated with probiotic use. When using live interventions, it is imperative that unit guidelines are designed with input from local microbiology teams so that patient and product safety can be monitored [20]. If they are not, adverse events associated with probiotic use could be under-reported in the literature'.